# Process evaluation of the BASIL+ trial: A behavioural activation approach to addressing low mood and depression among older people with long-term conditions

**Elizabeth Newbronner**[1]*, **Kate Bosanquet**[1], **Peter Coventry**[1,2], **Leanne Shearsmith**[3], **Elizabeth Littlewood**[1,4], **Della Bailey**[1], **Andrew Henry**[5], **Lauren Burke**[6], **Eloise Ryde**[1,7], **Dean McMillan**[1,8], **David Ekers**[1,4], **Simon Gilbody**[1,8], **Carolyn Chew-Graham**[9]

**1** Department of Health Sciences, University of York, York, United Kingdom, **2** York Environmental Sustainability Institute, University of York, York, United Kingdom, **3** Leeds Institute of Health Sciences, University of Leeds, Leeds, United Kingdom, **4** Tees Esk and Wear Valley NHS Foundation Trust, Research & Development, Flatts Lane Centre, Middlesbrough, United Kingdom, **5** Homerton Healthcare NHS Foundation Trust, Hackney, London, United Kingdom, **6** Manchester Institute of Education, University of Manchester, Manchester, United Kingdom, **7** Bradford Institute for Health Research, Improvement Academy, Yorkshire & Humber ARC, Bradford, United Kingdom, **8** Hull York Medical School, University of York, Heslington, York, United Kingdom, **9** School of Medicine, Keele University, Staffordshire, United Kingdom

* liz.newbronner@york.ac.uk

## Abstract

Older adults are at risk of low mood and depression, which can be exacerbated by long-term physical health conditions, leading to poorer health outcomes and increased mortality. The restrictions on social mixing during the COVID-19 pandemic heightened these risks. BASIL+ (Behavioural Activation in Social IsoLation) was a pragmatic randomised controlled trial conducted with patients recruited from general practices in England and Wales. It was designed to assess the effectiveness of behavioural activation in mitigating depression and loneliness among older people during the COVID-19 pandemic. A behavioural activation intervention, within a collaborative care framework, was delivered by telephone. Participants were offered up to eight weekly sessions with trained BASIL+ Support Workers. A qualitative process evaluation (conducted within the BASIL+ trial), used one-to-one semi-structured interviews to explore the experiences of trial participants, caregivers and BASIL Support Worker experiences. Three main themes emerged from the data analysis: 1) Engagement Dynamic: from Altruism to Self-Realisation; 2) Changing Behaviour and Improving Mood; and 3) Synergistic Nature of the Intervention Components. Findings suggest that the intervention effectively fostered self-awareness among older adults, giving them the confidence and skills to address low mood. Participant engagement with intervention materials varied, highlighting the importance of flexibility in delivery. BASIL+ Support Workers acted as key facilitators, guiding participants through the intervention process, fostering understanding, and providing crucial support. The collaborative care element ensured participants received comprehensive and holistic support, promoting improved mental and physical well-being. These findings underscore the adaptability and flexibility of the BASIL+ intervention, demonstrating its potential to address mental and physical health problems among individuals with varying

**Data Availability Statement:** The data from this study contains potentially sensitive information, and making it available without restriction would not be in accordance with ethical approvals. However, anonymised data will be made available upon reasonable request, which must include a protocol and analysis plan and not be in conflict with our prespecified publication plan, consistent with our data sharing policy (available on request via email from SG). Requests for data sharing will be considered by SG and the independent trial steering and data monitoring committee. For data sharing requests contact Pete Mather (pete. mather@york.ac.uk), Faculty IT Team Leader (Health Sciences).

**Funding:** The research was funded by National Institute for Health Research (NIHR) Programme Grants for Applied Research (PGfAR) RP-PG-0217-20006, and the grant holder are Prof David Ekers and Prof Simon Gilbody. The funder had no role in study design, data collection and analysis, decision to publish, or preparation of the manuscript.

**Competing interests:** SG and DE are members of the NICE Depression Guideline (update) Development Group. CAC-G declares royalties from Cambridge University Press Primary Care Mental Health, honoraria as Editor-in-Chief of Health Expectations and for organisation of the Royal College of General Practitioners (RCGP) One Day Essential learning event, and is Chair of the Society for Academic Primary Care awards panel and the RCGP Research Paper of the Year. All other authors declare no competing interests. This does not alter our adherence to PLOS ONE policies or data sharing and materials.

needs and different starting points. The BASIL+ trial is registered with the ISRCTN registry (ISRCTN63034289).

## Introduction

Aspects of aging, including retirement, development of long-term health conditions, functional decline and bereavement, place older adults at risk of low mood and depression [1]. Such life changes can also lead to older people experiencing increased social isolation and loneliness [2–4]. Moreover, comorbid depression in older people with long-term physical health conditions can worsen health outcomes and increase mortality [5,6]. During the COVID-19 pandemic, these risks were heightened when restrictions on social mixing disrupted peoples' daily routines and reduced social contact. Behavioural Activation (BA) is an evidence-based psychological treatment that explores how physical inactivity and low mood are linked and can result in a loss of or reduction in valued activities [7,8]. It is known to be effective in treating depression in older adults [9,10]. At the start of the COVID-19 pandemic in March 2020, an ongoing programme of work into the role of BA in older adults with multiple long-term conditions (multimorbidity) was adapted to answer the following overarching question: can a brief behavioural intervention delivered via telephone mitigate depression and loneliness in at-risk older people during COVID-19 restrictions? The adaptation involved two main changes– moving from face to face to telephone support for the older adult participants and moving training and support for the Support Workers to online/telephone. The resulting BASIL+ (Behavioural Activation in Social IsoLation) programme of work included several phases of research, indicative in the development of a complex intervention[11]: an initial pilot trial [12] with a nested qualitative interview study [13], followed by a pragmatic randomised controlled trial [14], and an embedded qualitative process evaluation–the focus of this paper.

The BASIL+ (Behavioural Activation in Social IsoLation) trial was a pragmatic full scale randomised controlled trial (ISRCTN registry reference ISRCTN63034289), conducted among patients recruited from general practices in England and Wales. Eligible participants were aged 65 years and older, with two or more long term conditions [9] and symptoms of low mood or depression (indicated by a score of 5 or higher on the Patient Health Questionnaire-9 (PHQ-9)) [8]. Participants were recruited from February 2021 to February 2022, a period during which shielding and social distancing measures formed part of COVID-19 restrictions [10]. Due to these restrictions, all the trial activities had to be delivered remotely. Early evidence from the BASIL+ pilot trial, which delivered BA remotely, suggested that this was both feasible and acceptable [7,11]. More recently, results from the BASIL+ main trial have shown that BA is an effective and potentially scalable intervention that reduces depression and some aspects of loneliness in older adults. [9].

This paper reports findings from the qualitative process evaluation within the BASIL+ trial [9], which was designed to explore trial participant, unpaid caregiver and BASIL Support Worker experiences of the BASIL+ intervention and enhance understanding about how older adults engaged with and used BA techniques, delivered remotely during the COVID-19 pandemic, to improve their mood and reduce loneliness. We explored how such an intervention might be scaled up and delivered within the NHS, voluntary or community organisations.

## Methods

### BASIL⁺ intervention

The BASIL⁺ intervention was based on the principles of BA which aim to help people maintain or reinstate activities (or replace them if they are no longer possible) that are important to them, and, which benefit their physical and emotional wellbeing by helping them stay socially connected. The intervention comprised the offer of up to eight sessions delivered remotely by a trained Basil Support Worker (BSW). The sessions were supported by a self-help booklet which begins by exploring *"The link between what you do and how you feel"* and is structured around the eight sessions. These elements were set within a 'light touch' collaborative care framework whereby BSWs could liaise, with participants' consent, with other professionals relevant to the participant's health and social care needs (e.g. to facilitate obtaining appointments or reviews) or with voluntary and community organisations. However, they did not engage in structured on-going activity with participants health and care professionals in line with the traditional collaborative care framework [7]. BSWs also sign-posted participants and provided them with information, encouraging them to be pro-active in seeking support. BSWs participated in an online bespoke training course which was delivered by mental health professionals with expertise in BA. BSW backgrounds included community nurses and support workers, psychological wellbeing practitioners (typically employed in the English NHS Talking Therapies services), assistant psychologists, care coordinators, and physiotherapists. More information about the BASIL+ intervention and full details of the methods used in the whole trial, including recruitment timeframes and processes, and arrangement for obtaining informed consent, can be found in the trial papers [12–15].

We conducted a qualitative process evaluation, which used semi-structured interviews to explore the acceptability and feasibility of the intervention, and how it might be implemented, from the perspectives of older adult participants, caregivers/supportive others (where they have been involved in the intervention), and BASIL Support Workers.

Ethical approval (IRAS: 293203) was granted by Yorkshire & The Humber - Leeds West Research Ethics Committee on 11/12/2020. The BASIL⁺ trial was informed by a Patient and Public Involvement Advisory Group (PPI AG) that had been established to inform the pre-COVID research programme and continued throughout the BASIL⁺ trial. This PPI AG included older adults with lived experience of mental health or physical health conditions as well as caregivers.

### Participant selection

**Older adults.** BASIL⁺ trial participants provided their optional consent to take part in a qualitative interview at the point they joined the trial. From those consenting participants who were randomised to the BA intervention group, and who completed the intervention, a purposive sample was selected based on four characteristics:

- Gender

- Age (65 to 79 and 80+ years)

- PHQ-9 score of 5 or above [16]
  *(5 to 9 classified as mild depression; 10 to 14 moderate depression; 15–19 moderately severe depression; ≥ 20 as severe depression)*

- Type of long-term condition/s recorded at trial entry (see Table 1)

- Ethnicity

**Table 1. Older adult 'completer' participant characteristics.**

| Characteristic | Number |
|---|---|
| Gender: | |
| Female | 11 |
| Male | 10 |
| Age Group (years): | |
| 65–79 | 13 |
| 80 and over | 8 |
| PHQ-9 score: | |
| 5–9 | 10 |
| 10 and over | 11 |
| Type of Long-Term Condition: (people disclosed more than one condition) | |
| Arthritis (e.g. rheumatoid arthritis, osteoarthritis) | 6 |
| Diabetes | 7 |
| Cancer | 1 |
| Cardiovascular conditions (e.g. heart problems, heart attack, heart failure, hypertension, atrial fibrillation) | 13 |
| Chronic Pain | 2 |
| Neurological conditions (e.g. Epilepsy, Parkinson's Disease, Multiple Sclerosis) | 3 |
| Osteoporosis | 2 |
| Respiratory Conditions (e.g. COPD, asthma) | 4 |
| Other | 6 |

The older adults were drawn from the eleven sites which recruited participants to the trial. They were contacted by a researcher to confirm they were still willing to take part in a qualitative interview once they had completed their three-month follow-up questionnaire. We also sought to include participants who had consented to take part in the qualitative study and who remained in the trial but who did not complete the intervention, so we could explore their reasons for not continuing.

**Caregivers/Supportive others.** Caregivers were defined as partners, family members or friends who provided practical and/or emotional support to the older adult. BSWs were asked to identify caregivers who had played a role (e.g. joining in some of the sessions) in supporting the older adults to take part in the intervention. The BSWs then obtained verbal permission from interested caregivers/supportive others for the BASIL[+] study team to send them a study information pack (containing an invitation letter, participant information sheet and example consent form). This was followed by telephone contact from the study team to discuss the study and, if they agreed, to arrange an interview. It was anticipated that caregivers/supportive others may be a difficult participant group to identify based on findings from the BASIL pilot trial. No pre-defined sampling was employed, and all those who agreed to take part were interviewed.

**BASIL Support Workers (BSWs).** Upon completion of their training, all BSW were sent a study information pack (containing a letter inviting them to take part in the process evaluation, a participant information sheet and an example consent form). They were asked to complete an online consent form or to contact the qualitative study team directly to indicate their interest in taking part in an interview. The main factor considered when selecting BSWs was the recruiting site they worked in. However, efforts were also made to include a mix of genders and ethnic backgrounds and to take account of the length of time they had been a BSW.

## Data collection

One-to-one semi-structured interviews were conducted by telephone. Separate topic guides were developed for each group of participants i.e. older adult participants who completed the intervention ('completers'); those who did not complete the intervention ('non-completers); caregivers/supportive others; and BSWs. However, all the topic guides explored the acceptability of the intervention, including important elements of the intervention and mode of delivery, barriers and enablers to integrating the intervention into participants' existing health and care support, and the impact of the intervention on participants' mood and general wellbeing in the context of the COVID-19 restrictions. For BSWs, additional questions explored barriers and facilitators to delivery of the intervention and how the intervention might be integrated within health and care settings. The topic guides for older adult participants and caregivers/supportive others were discussed and piloted with members of the BASIL+ PPI AG. The topic guide for the BSWs was piloted with two BSWs who had taken part in the BASIL pilot trial (these BSWs were not selected for interview for the process evaluation). Following feedback from both groups, the final topic guides were produced.

The interviews with older adult participants and caregivers/supportive others took place between July 2021 and April 2022. Interviews with BSWs were conducted between October 2021 and February 2022, allowing time for BSWs to have supported more than one participant. The interviews were conducted by three researchers (EN, LS and AH). They were digitally recorded (with participant consent) using an encrypted digital recorder and transcribed verbatim by a professional transcribing company. The transcripts were then checked and anonymised by the interviewer. Transcripts were not returned to participants.

## Data analysis

The anonymised transcripts were uploaded to NVivo 12 software for analysis. A deductive and inductive approach was adopted in that the concepts from the interview topic guides were used to provide an initial outline framework for the analysis (i.e. deductive). Within this we then used inductive thematic analysis in which themes and codes emerged from the data [17]. Three researchers (EN, KB and LS) conducted the initial analysis, with ongoing discussion about the coding with two more experienced members of the team (PC and CCG). The qualitative study team then discussed revisions to the coding structure before analysis proceeded. The qualitative study team met regularly to discuss emerging themes, develop findings, and approve the results.

## Results

Forty-two interviews were conducted across 11 sites in England and Wales: 21 'completer' interviews; three 'non-completer' interviews; two caregiver interviews and 16 interviews with BSWs. Table 1 provides the key characteristics of 'completer' participants and demonstrates a relatively even mix of participant characteristics based on the variables purposively sampled: gender, age, PHQ-9 score and type of long-term condition. Just one participant self-identified as being from a minority ethnic group. The three 'non-completers' participants included two women and one man, two aged between 65 and 79 years and one over 80 years. The BSWs were drawn from 10 sites and included support workers who had taken part in the BASIL pilot trial as well as those who were newly trained for the BASIL+ trial. The 'completer' interviews lasted between 20 and 60 minutes, and BSW interviews between 20 and 80 minutes. The 'non-completer' and caregiver interviews were typically much shorter, generally lasting between 20 and 30 minutes.

## Context of the findings

The BASIL[+] intervention was designed to support older people with existing long-term conditions (LTCs) who were experiencing low mood and who were socially isolated as a result of COVID-19 restrictions. However, it was clear from the qualitative study data that for many participants the underlying reasons for their low mood or social isolation pre-dated the pandemic restrictions, and so the intervention was often addressing wider concerns. For some participants, their low mood was as much related to wider difficulties or changes in their lives, as it was to their long-term health conditions or the pandemic restrictions. Bereavement or relationship problems, usually with their partner but also with grown-up children, were often described as being responsible for loneliness. Our analysis therefore suggested that for most participants the COVID-19 pandemic exacerbated existing social isolation and loneliness rather than caused it.

Findings from the interview data address three core themes, each of which is discussed in depth in the sections below:

- Engagement Dynamic: from Altruism to Self-Realisation

- Changing Behaviour and Improving Mood

- Synergistic Nature of the Intervention Components

Illustrative data are provided to support our analysis. They are labelled according to the type of participant (i.e. 'Comp' for Completer; 'Non-Comp' for non-completers; and 'BSW' for Basil Support Workers) and ID code.

## Engagement dynamics: From altruism to self-realisation

Most participants engaged with the BASIL[+] intervention despite some having talked about how altruism was their initial motive for taking part in the trial. These participants viewed taking part in a research study as an opportunity to help others, over and above seeing it as an opportunity to help themselves. As one older adult explained: *"well I didn't think about myself. I was just willing . . . in the hope that it may help somebody else. I'm not really a depressive person"* (Comp-16007). Several indicated a similar desire to help others and do something for the greater good: *"I did it to . . . anything that can improve, help, for other people it's a worthwhile thing to do"* (Comp-14011).

> *I hoped it would throw light on the difficulties that had been for the elderly, especially, trying to get hold of the doctor's services . . . throw light, how things could have been dealt with perhaps better than they were*. Comp-15006

Disclosed motives for signing up to and engaging with the BASIL[+] intervention varied. Whilst the majority who reported altruistic motives were not initially primed to engage with the intervention because of their health or sense of isolation, they nonetheless completed the intervention. The non-completer participants described struggling to engage with the study because they did not believe themselves to be depressed and therefore thought the intervention was not appropriate for, or relevant to them.

> *I had no expectations whatsoever, I was just curious to see what it was. Once we got going I realised, well you might be able to help some people but with the problem I've got no amount of talking to you people will ease it off*. Non-Comp-10111

*At the moment my life is perfectly fine but who knows in a few years time, and if I can chip in and it can help, and other people.* Non-Comp 19041

*There was talk about being depressed. . .I don't think I've ever been depressed. I don't really know what it means. . .I don't have low mood either.* Non-Comp-11006

Mirroring this, most BSWs felt that for participants to respond to the BASIL⁺ intervention, their mood needed to be low enough for them to want help. Some BSWs were concerned their participants were not experiencing low mood and so their capacity to engage with the intervention was limited. As one BSW remarked, *"I suppose some of* [my participants] *haven't had as low mood as you think, that they've been too well, I suppose, maybe for the intervention"* (BSW-111). Another BSW reflected: *"there's probably something around their mood, needs to be low enough for them to feel like they need it"* (BSW-102).

In some instances where participants initially harboured doubts, either because they had *". . .never been troubled by any mood, mental or depression things, ever"* (Comp-10119) or were undeserving of attention, they still engaged with the intervention, finding it "useful" and "beneficial". Similarly, others experienced a shift in perception after giving the intervention a chance, illustrating that engaging in the intervention surfaced challenges that they had not fully appreciated:

*"I think I was in denial until I did the initial assessment, whether I was a suitable subject or not [for the intervention]. That kind of alerted me that in fact I wasn't managing nearly as well as I'd like to think I was"* (Comp-11010).

The interviews with older adult participants and BSWs demonstrate that, even when there were concerns about whether an older adult was 'eligible' or 'felt deserving' of the intervention, if they were willing to try it, most went on to complete the intervention and felt that they had benefitted from it.

## Changing behaviour and improving mood

Many participants talked about seeing a change in themselves or described some improvement in their mood, which they attributed to taking part in BASIL⁺. One participant who at the outset had not recognised how low he was said: *"I am more honest about my mood. . .I feel good, you know, when I can achieve something. . .So, yes, having some fairly modest goals and being able to achieve some of them, anyway, has made a big difference* [to] *my mood" (*Comp-11010). Several participants and BSWs credited BASIL⁺ with having motivated them:

*If you* [BSW] *hadn't have phoned me and asked me to take part in it, I think I'd have still been a cabbage sat on the settee. It did fetch me out of being in a cocoon, like being bored and everything.* Comp-11017

For most participants any improvement described was often more modest, describing a change in thinking or mindset, or that they felt more positive. When asked about any change in mood, one participant said: *"I'm sure it's improved. You know, it makes me think a little bit more and be positive"* (Comp-17109).

Both BSWs and older adult participants said that for people to really see a change in their mood, they had to be experiencing low mood or depression at the start of the intervention. A small number of older adults said that whilst they had enjoyed taking part in BASIL⁺, they did not think it had made a discernible difference to their mood. Others talked about learning

strategies for lifting mood when they felt a bit low. One participant said they understood how small actions could change their mood saying: *"If your mood is very low, try and distract your-self, look for distractions, take up something to work on"* (Comp-12046). Others talked about having a better understanding of how to look after their psychological wellbeing, even if they were generally feeling well.

Several participants said that the process of talking to their BSW about how they felt, helped them talk more openly to others: *"It was strange at the beginning sort of being able to. . .how can I say this. . .err come out, speak out, to say things that I would probably bottle it up inside. It made us be able to. . .talk about it"* (Comp-12001). Similarly, others felt that taking part in the intervention had given them the language to talk about how they were feeling, and this made it easier for them to talk to family and friends about it: *"To be quite honest, I think, talking to* [the BSW] *and doing this thing, it's made me a bit more laid back. . .I've been slightly more open with my daughter as well. . .you know, by talking to her about different things"* (Comp-14000).

The extent and nature of the changes made by participants because of the intervention varied. Several participants described how taking part in BASIL$^+$ had led them to make quite substantial changes. For some this involved returning to activities they had enjoyed in the past or taking up new activities or taking active steps to be more sociable:

*One of the things I've done is got a few people involved to go and play bowling in the park near us. My next door neighbour, he's an elderly gentleman, he goes with me, and then I got one or two other people involved as well, so there's a group of us we have started going out. And the other thing, you know, to be more socially active, so I've started going out to the pub once a week just for a short time for a couple of drinks. It has helped.* Comp-15018

Other older adult participants talked about how the intervention had helped them understand the importance of having structure or routine in their lives or enabled them to regain a routine after the disruption of illness or pandemic restrictions. One participant disclosed that he had become quite demotivated, and his personal self-care had declined. With the support of his BSW he began actively planning his week and take up new activities: *"She* [BSW] *taught me how to be better structured. I didn't think I needed teaching at my age, but I did".* Comp-10057. Conversely there were a few participants who needed to allow themselves more flexibility:

*It was mainly just the need not to have everything done in a routine. I was always the person that everything had to be done in a routine, and it made me realise that life would be a bit eas-ier if I was more flexible. . . I honestly no longer feel I've got to rush around and get everything done. It has helped. It really has.* Comp-10119

For most participants, whilst the changes made were often small or subtle they were none-theless important, especially where they made people feel they were progressing or gave them a greater sense of self-reliance. Older adult participants often gave examples of small things they had done that made them feel positive:

*I'd tell her [BSW] I put a shoe rack together, because although it might seem nothing but I had a husband that did everything. All I had to do was look after him, full stop. I didn't do any-thing else. Anything that needed doing in the home, whatever it was, he did it, and for me, that was a big achievement for me, and ordering furniture, well, I'd never done that in my life, you know?* Comp-16007

The BSWs observed that it was sometimes difficult to see what impact the intervention was having on their participants. This was partly because changes could be incremental and subtle, but also because participants did not always articulate the changes they were making. The BSWs also felt that it was important not to make assumptions or anticipate what was going to be significant for participants.

## Synergistic nature of the intervention components

As described, the BASIL$^+$ intervention had three components: Behavioural Activation (a brief structured psychosocial intervention tailored to older adults), the main concepts and tools of which were set out in the self-help booklet; telephone support from a trained BSW; and 'light touch' collaborative care (e.g. the provision of information, signposting to other sources of care or support liaison with the participant's GP practice).

Whilst some older adult participants reported finding the concepts like the BA cycle (described in the self-help booklet) too abstract, others felt that, together with more practical tools, (e.g. diary exercise and scheduling activities), helped them to see how changing their behaviour could change how they felt. BSWs claimed that they played a key role in helping participants make this connection and motivating them to change. As one older adult participant explained:

> *Like I say, she got me into doing things instead of just sitting around watching TV and being stuck not being able to go anywhere. . .I had a problems with me shed and she got us motivated to getting it sorted out and getting rubbish out of me garden and yeh she got us motivated into doing things. Comp-12001*

The BA component of the BASIL+ intervention is, by its nature, a structured approach to tackling low mood. It is organised around eight supported BA sessions, which also provided the format for the participant booklet. Most BSWs reported that the booklet gave them a helpful structure for their sessions with participants. One BSW commented that "[the booklet] *was really easy and a really good guide for me actually, as to what we had to go through. . . every session I'd have that open and work through it with the participant"* (BSW-112). However, they also felt they needed to be flexible in following the structure depending on the participants' needs. They reflected that taking a flexible approach kept the approach participant-centred and helped to keep older adults engaged by focusing on sections of the booklet most relevant to them:

> "*I think the flexibility of being able to not necessarily cover every single session if it isn't relevant or recap sessions or to skip sessions and that is obviously important because otherwise it's not then patient-centred is it, you're sort of almost restricting them to working through something that might not be relevant to them. So I think the fact that you can make those decisions with them and decide what is relevant helps.*" BSW-106

However, some BSWs commented that taking this more flexible approach involved additional preparation prior to the sessions with participants, and a level of confidence and familiarity with the booklet and BA materials.

The reported level of participant engagement with the booklet varied. Some older adults described reading the whole booklet right at the start of the intervention, whilst others used it to prepare for each session, often writing notes: "*I used to read the section and then I'd write notes on it so that when she asked me I could read back things that I'd written on it. . .and then we used to discuss that"* (Comp-16007). Several older adults saw it as a resource that they could keep and go back to in the future:

*"I've still got the booklet and I think if I, if I felt, if I went back to, if I felt low or anything, I might well look at it again, yes. . .if I suddenly went through a phase of feeling a bit down, I think I would look at it again. Yeah. I mean, I wouldn't get rid of it."* - Comp-19050

However, other older adults suggested that they only engaged with the booklet as part of the conversations with their BSW. In many ways this group of participants highlight the value of combining BA with support from a trained support worker.

Many participants described the value of having someone different to talk to during the COVID-19 pandemic restrictions, who was not a family member or healthcare professional (HCP), and unlikely to judge them.

*I think for some of them it was particularly when they weren't getting out much or seeing many other people, and it was just important to have that source of support. And kind of again, a regular contact and, maybe, be able to speak a little bit more openly with someone who's outside of. . .you know, there's a lot of, kind of, feeling of being a burden, so they might not have wanted to, sort of, burden family members with how they were feeling and things.* BSW-104

It was the structured nature of these conversations, and the fact that they were taking place in the context of the intervention that seemed to make the difference, allowing older adults to reflect on their situation, to open up about difficult things such as feeling low or lonely, or to think through difficulties in a contained way.

*"Let me try to phrase it right. When you're down, just talking to somebody makes you focus and look at things slightly differently. So, yes, that has definitely happened. Generally, I have been quite a motivated person in life, and I don't give up very easily, but this has definitely, sort of, I got from that, you know, don't stay in and do something about it."* Comp-15018

An important feature of the intervention for many participants was its collaborative nature. The degree of collaboration with other professionals ranged from quite significant involvement, including the BSW making direct contact with the participants' healthcare practitioners, to sign-posting and information-giving. A number of BSWs reported phoning or writing to the older adult's general practitioner (GP) or Practice Nurse to ask them to contact the participant, or to request an appointment on their behalf. This was often where participants were finding it hard to get an appointment because of pressure on services or were reluctant to contact their GP. In one area, some of the BSWs were also Health Care Assistants in the general practice where the older adult was registered, and they were able to make an appointment for the older adult with another clinician in the practice.

There were a few participants who were consulting their GP or Practice Nurse for their physical health problems but who had not had their mental health needs addressed. One participant explained that although he had seen his GP for a check-up and blood tests, it was only after the BSW had spoken to the GP that his depression was recognised and treated: *"She has spoken to my GP with regard to that and I feel that was the reason that I went onto the Fluoxetine."* (Comp-18007).

For a small number of participants, the impact on their ability to manage their physical health condition was reported to be more important than any change in their mood. As one BSW explained:

*"They sort of felt that they were okay. . . but actually, through just sort of talking about the different sessions that were in the booklet, we were able to identify various things each session that helped them to move forward with managing their long-term condition. So, like, you know, just being a bit more proactive at speaking to the GP if they're unsure about their medication or requesting a review from their doctor because they weren't sure what the outcome was of some tests they'd had done".* BSW-106

BSWs also described referring people to other health services; one referred a participant to the local memory clinic. Another explained: *"There was another woman, she was quite poorly sighted and she kept dropping her tablets and stuff . . .I ended up organising an OT to go out and assess her and put some adaptations in place."* (BSW105).

The collaborative care aspects of the intervention were 'light touch', such as sign-posting to other services for support or information. Most of the BSWs interviewed had signposted their participants to sources of support and information, often from Third sector organisations and/or charities supporting people with their condition. One BSW said: "*I contacted the charity first just to check what they did and whether she could ring them and how they could help her and signposted her there to get more information about the condition."* (BSW 102). Others directed participants to useful websites or, where they didn't have access to the internet, they posted hard copies of information to them. One older adult who had recently been diagnosed with Crohn's Disease explained: *"She got me a few pages on diet and foods to eat and food to avoid and things like that and it made me look up on the computer, on what do they call it, the internet, the web".* He went on to say: *"I'd never looked medical things up, but I certainly looked at this–Crohn's Disease"* (Comp-12001).

Some BSWs reported that they had sign-posted people to groups or activities in their communities and, occasionally online, simply to help them become more active or socially connected. Some older adults reported following the suggestions up, but others said they were still thinking about it, in some cases waiting for the COVID-19 pandemic restrictions to be lifted. A few older adults said that they were not going to follow the suggestions up, either because they lacked motivation or did not like the suggested activities.

Our analysis suggests that whilst response to each intervention component varied between participants, they were nevertheless synergistic, with BSWs proving pivotal to participants' understanding of BA, making the booklet more relevant, and adopting a collaborative where appropriate and feasible.

## Discussion

The process evaluation revealed key findings about how the BASIL[+] intervention functioned for participants and BSWs in the context of periods of enforced isolation during the COVID-19 pandemic. Firstly, even where the BASIL[+] intervention was initially perceived to be not relevant, older adults attributed self-realisation and personal growth to engagement with the intervention. This transition reflects the intervention's effectiveness in fostering self-awareness among older adults and giving them the confidence and skills to engineer more healthy ways to live. Secondly, the varying levels of participant engagement with intervention materials highlight the importance of flexibility in delivery, allowing for individualised support aligned with participant preferences and needs [18]. Thirdly, BSWs emerged as key facilitators, guiding participants through the intervention process, fostering understanding, and providing crucial support. Their adaptability significantly contributed to intervention success. Lastly, the collaborative care approach, involving coordination with healthcare professionals, information giving and sign-posting to other support services, ensured participants received

comprehensive and holistic support, promoting improved mental and physical well-being. Overall, these findings underscore the adaptability and flexibility of the BASIL+ intervention, demonstrating how it can be deployed to address a range of problems among people whose starting points differ.

The transformation among those that initially did not see the intervention as relevant signalled that the BASIL[+] intervention has potentially preventive qualities, capable of supporting people to proactively maintain well-being and mitigate age-related health decline. Moreover, BASIL[+] arguably reached some people who may not have otherwise sought help or who may not have been eligible to receive mental health support. As such, the BASIL[+] intervention offered older adults personalised opportunities to engage in a structured and therapeutic intervention not ordinarily accessible through existing adult mental health services, and so address problems in a manner that suited them. Having the potential to tailor the intervention to suit the participant's preferences and circumstances proved crucial to meeting the needs of individuals. This highlights the intervention's flexibility in supporting individuals with diverse social and psychological needs [18] enabling them and support workers to deconstruct the intervention components and draw upon those that meet their needs. Furthermore, we found the components of the intervention were synergistic and worked in ways that complemented each other.

Our analysis echoed the main quantitative findings for the primary outcome [13], telling a story of a reduction in depressive or low mood symptoms in response to the intervention. We had anticipated that participants may have felt more lonely than normal during the COVID-19 pandemic, as shown in other studies conducted during the same time period [19], but this narrative did not emerge from the interviews.

Overall, the findings demonstrate a core strength of the BASIL[+] intervention was its complexity, a structured approach combined with flexibility in its delivery [18], which meant it could be person-centred and tailored to an individual's needs [20,21]. In most cases, the intervention had a positive impact on mood and wellbeing which many participants credited to small, subtle, incremental changes they had made to their behaviour or daily routine.

## Limitations

The study has several limitations. First, nearly all participants self-identified as White British, meaning there was limited ethnic diversity. For this reason some caution may be appropriate when generalising our findings to the wider and more diverse populations [22]. Another limitation is the sample may have been subject to selection bias as participation in the process evaluation was voluntary, and, it is possible that people who chose to take part differ from those who did not [23]. Furthermore, because the study took place during the COVID-19 pandemic where the context was different, there was an urgent need to recruit and train BSWs quickly, so for logistical reasons it was challenging to test the feasibility of people delivering the intervention who were working in their usual patient-facing NHS or voluntary sector roles. In most of the study sites, the intervention was delivered by BSWs who were drawn from existing staff already working in NHS Trust Research and Development (R&D) teams. However, one site enabled Health Care Assistants working in a GP Practice to be BSWs and another provided the opportunity for Community Support Workers employed by a Voluntary and Community Service organisation to be BSWs.

## Implications for practice and research

The BASIL[+] qualitative findings have provided insights into how this type of intervention could be delivered within existing mainstream NHS and voluntary and community sector services. It has shown the intervention is flexible and it can be delivered in a person-centred

manner. In this sense it has the potential to be offered as an alternative pathway to existing NHS Talking Therapies services [21,24]. It could, therefore, potentially fill a gap in mental health services.

We suggest that non-mental health staff in existing health and care roles (e.g. Social Prescribers, Health Coaches, practitioners in voluntary and community organisations) could be trained to deliver the BASIL[+] intervention. In this context it should be seen as one of the 'tools in their toolkit' which they can use to support older adults with low mood, rather than a separate service. Moreover, training them in this brief psychological intervention has the potential to provide them with skills and knowledge which they can use to inform their work more generally, even when they are not delivering the full intervention. Effectively, being trained to deliver BA would upskill professionals working with older adults and have a positive trickledown effect on their routine work. We identified several practical issues that would need to be addressed to ensure a BASIL[+] type complex intervention could be delivered safely and appropriately.

Furthermore, it is important to underline that in addition to the intervention being complex and consisting of numerous components, it was delivered within a context of a healthcare system under pressure due to the COVID pandemic and funding constraints. As such, the implications of this process evaluation signal the need to move towards the next phase of research and explore the process of implementation in the real system where its feasibility is tested in real world settings at several sites.

Potential implementation challenges for consideration are outlined below, although we recognise that more research is needed to test the approach we are suggesting in real world settings in the NHS and voluntary and community organisations:

- Arrangements for case-finding 'at risk' groups to identify people for suitability for such an intervention

- Protocols/referral routes for someone who presents with more complex needs

- Training staff/practitioners to deliver the intervention

- Supervision arrangements for staff/practitioners delivering the intervention

- Guidance about caseloads/how many people a worker can be expected to support through the intervention at any one time.

## Conclusion

This qualitative process valuation conducted as part of the BASIL[+] trial has enhanced understanding about how older adults engage with and use behavioural activation techniques, delivered remotely during the COVID-19 pandemic, to improve their mood and reduce loneliness. We suggest the intervention could also be used as a tool to help older adults stay well. How the BASIL+ intervention fits in with current service configurations, given its complexity, and how it might be commissioned and funded needs further investigation.

## Acknowledgments

We would like to thank: all the older people, caregivers and Basil Support Workers who took part in the process evaluation; the participating NHS Trusts, general practices and Age UK organisations for their support; and the Ethics Committee for overseeing the study. We are also grateful to our PPI AG members for their insightful contributions and collaboration.

## Author Contributions

**Conceptualization:** Dean McMillan, David Ekers, Simon Gilbody, Carolyn Chew-Graham.

**Data curation:** Elizabeth Newbronner, Kate Bosanquet, Elizabeth Littlewood, Lauren Burke, Eloise Ryde.

**Formal analysis:** Elizabeth Newbronner, Kate Bosanquet, Leanne Shearsmith, Carolyn Chew-Graham.

**Funding acquisition:** David Ekers, Simon Gilbody.

**Investigation:** Elizabeth Newbronner, Leanne Shearsmith, Andrew Henry.

**Methodology:** Elizabeth Newbronner, Peter Coventry, Elizabeth Littlewood, Della Bailey, Carolyn Chew-Graham.

**Project administration:** Elizabeth Newbronner, Elizabeth Littlewood, Lauren Burke, Eloise Ryde.

**Supervision:** Peter Coventry, Carolyn Chew-Graham.

**Writing – original draft:** Elizabeth Newbronner, Kate Bosanquet, Leanne Shearsmith.

**Writing – review & editing:** Elizabeth Newbronner, Kate Bosanquet, Peter Coventry, Elizabeth Littlewood, Della Bailey, Carolyn Chew-Graham.

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
