## [Decision Letter · Decision Letter 0]

9 Sep 2024

PMEN-D-24-00283

Process Evaluation of the BASIL+ trial: a Behavioural Activation approach to addressing low mood and depression among older people with long-term conditions

PLOS Mental Health

Dear Dr. Newbronner,

Thank you for submitting your manuscript to PLOS Mental Health. After careful consideration, we feel that it has merit but does not fully meet PLOS Mental Health’s publication criteria as it currently stands. Therefore, we invite you to submit a revised version of the manuscript that addresses the points raised during the review process.

A rebuttal letter that responds to each point raised by the editor and two reviewer(s). You should upload this letter as a separate file labeled 'Response to Editor and Reviewers'.A marked-up copy of your manuscript that highlights changes made to the original version. You should upload this as a separate file labeled 'Revised Manuscript with Track Changes'.An unmarked version of your revised paper without tracked changes. You should upload this as a separate file labeled 'Manuscript'.

We look forward to receiving your revised manuscript.

Kind regards,

Dr Gareth Hagger-Johnson

Academic Editor

PLOS Mental Health

Journal Requirements:

Additional Editor Comments:

The intervention is a complex intervention with at least three active ingredients / mechanisms.

Refer to the framework for complex intervention evaluation more explictly throughout.

https://www.bmj.com/content/374/bmj.n2061

in particular complex interventions as events in systems. Make more of the interaction between the intervention and its delivery context. e.g. does your evaluation have any implications for theory modification?

L130 Is purposive the right word?

L140 Completers vs non-completers should be a bullet here (and an entry in Table 1) with numbers

L195 Brief explain what deductive and induction means to readers unfamiliar with qualitative research

L230 Reference needed for existing quantitative research on pre-pandemic social isolation (e.g. introduction here https://www.frontiersin.org/journals/psychology/articles/10.3389/fpsyg.2020.02201/full)

L232 Please avoid 'speak to' which sounds fluffy and informal

L240 Should be non-completers not none-completers

L541 Please add references to quantitative findings, even if in press / recently accepted

L600 Not clear when you say BASIL+ qualitative study if you mean this paper or another qualitative paper?

Reviewers' comments:

Reviewer's Responses to Questions

**Comments to the Author**

1. Does this manuscript meet PLOS Mental Health’s publication criteria? Is the manuscript technically sound, and do the data support the conclusions? The manuscript must describe methodologically and ethically rigorous research with conclusions that are appropriately drawn based on the data presented.

Reviewer #1: Yes

Reviewer #2: Yes

2. Has the statistical analysis been performed appropriately and rigorously?

Reviewer #1: N/A

Reviewer #2: N/A

3. Have the authors made all data underlying the findings in their manuscript fully available (please refer to the Data Availability Statement at the start of the manuscript PDF file)?

Reviewer #1: Yes

Reviewer #2: Yes

4. Is the manuscript presented in an intelligible fashion and written in standard English?

Reviewer #1: Yes

Reviewer #2: Yes

5. Review Comments to the Author

Reviewer #1: I have very few comments on this paper. It is a rigorous evaluation of the process of the BASIL+ trial. However:

I would have liked something more about the 'adaptation' of the (L67) for the setting of COVID in this paper to frame the findings, and comment on whether any further adaptation was found to be necessary, or not.

(L102) some explanation of why the collaboration care intervention was considered to be 'light touch' would be helpful to readers unfamiliar with collaborative care.

(L229) the section on context of the findings is interesting, but some data to support the statement in the line sentence beginning 'Analysis suggested...' would be helpful.

(L337) were there any examples of the reality of lockdown impeding chosen social activities or how these were modified to suit the circumstances given that the potential need to do this was recognised in the protocol?

(L507) missing word

Finally were there any particular issues that arose in supervision of the BSWs given the potential difficulties arising in such a study during COVID? Supervisors were not interviewed and may have provided some useful insights.

Reviewer #2: Overall, this is a very clear and well written paper reporting a qualitative process evaluation. The introduction describes the background well, the methods provide a detailed description of the study process. The themes developed in the results section are interesting, succinct and clearly described. The discussion is appropriate and offers further contextualisation of the insights offered in the results. Consequently, I have little to suggest by way of improvements.

One area to consider perhaps is that, as written, it is a very positive report of the intervention, its use and its impact. Which left me wondering if there was more to be said regarding any participants who did not get on well with the intervention? We are told that in most cases the intervention had a positive impact. What happened where it did not? Were these the participant who had taken part primarily for altruistic reasons rather than low mood (we also find this in our research), the non-completers? Were there any completers who didn’t like it? I understand if not, but it would be useful just to cover this a little more.

One other minor aspect to expand on is in the analysis section. It says a deductive and inductive approach was adopted. Could the authors expand? In what way was the approach to analysis deductive, that wasn’t clear to me.

As it stands, this process analysis represents an important and useful contribution to this area, complementing the trial publications. Expanding on the above issues may further strengthen the paper.

6. PLOS authors have the option to publish the peer review history of their article (what does this mean?). If published, this will include your full peer review and any attached files.

**Do you want your identity to be public for this peer review?** For information about this choice, including consent withdrawal, please see our Privacy Policy.

Reviewer #1: No

Reviewer #2: **Yes: **Adam Geraghty

---

## [Decision Letter · Decision Letter 1]

10 Dec 2024

Process Evaluation of the BASIL+ trial: a Behavioural Activation approach to addressing low mood and depression among older people with long-term conditions

PMEN-D-24-00283R1

Dear Dr Newbronner,

We are pleased to inform you that your manuscript 'Process Evaluation of the BASIL+ trial: a Behavioural Activation approach to addressing low mood and depression among older people with long-term conditions' has been provisionally accepted for publication in PLOS Mental Health. Both reviewers were happy with the changes made, and I am happy to accept this on a provisional basis.

Best regards,

Dr Gareth Hagger-Johnson

Academic Editor

PLOS Mental Health